# Olfactory Learning Supports an Adaptive Sugar-Aversion Gustatory Phenotype in the German Cockroach

**DOI:** 10.3390/insects12080724

**Published:** 2021-08-13

**Authors:** Ayako Wada-Katsumata, Coby Schal

**Affiliations:** Department of Entomology and Plant Pathology, North Carolina State University, Raleigh, NC 27695, USA

**Keywords:** gustation, sugar aversion, German cockroach, olfactory learning, memory, foraging

## Abstract

**Simple Summary:**

Toxic baits that contain an insecticide and phagosimulatory sugars, including glucose, are most effective in German cockroach control. However, cockroaches have evolved behavioral resistance, where they perceive glucose as a deterrent and avoid eating the bait (glucose-aversion, GA), resulting in failure to control infestations. We hypothesized that the GA phenotype may be extended by associative learning of specific odors with glucose. We demonstrated that GA cockroaches associated attractive food odors, such as vanilla and chocolate, with glucose (deterrent) and learned to avoid these odors. In contrast, wild type (WT) cockroaches that associated these odors with glucose (phagostimulant) increased their preference for the odors. The aversive and appetitive memories were retained for at least three days. Generally, when toxic baits are deployed, GA cockroaches are first attracted to the bait, and they repeatedly experience its aversive taste as they reject eating the deterrent bait. The recurring non-rewarding foraging experience may contribute to the formation of an aversive olfactory memory. Even if the baits are later reformulated without aversive tastants, GA cockroaches may avoid the new bait because they associate it with aversive olfactory stimuli. Our findings will guide the rational development of baits that consider the olfactory learning abilities of cockroaches.

**Abstract:**

An association of food sources with odors prominently guides foraging behavior in animals. To understand the interaction of olfactory memory and food preferences, we used glucose-averse (GA) German cockroaches. Multiple populations of cockroaches evolved a gustatory polymorphism where glucose is perceived as a deterrent and enables GA cockroaches to avoid eating glucose-containing toxic baits. Comparative behavioral analysis using an operant conditioning paradigm revealed that learning and memory guide foraging decisions. Cockroaches learned to associate specific food odors with fructose (phagostimulant, reward) within only a 1 h conditioning session, and with caffeine (deterrent, punishment) after only three 1 h conditioning sessions. Glucose acted as reward in wild type (WT) cockroaches, but GA cockroaches learned to avoid an innately attractive odor that was associated with glucose. Olfactory memory was retained for at least 3 days after three 1 h conditioning sessions. Our results reveal that specific tastants can serve as potent reward or punishment in olfactory associative learning, which reinforces gustatory food preferences. Olfactory learning, therefore, reinforces behavioral resistance of GA cockroaches to sugar-containing toxic baits. Cockroaches may also generalize their olfactory learning to baits that contain the same or similar attractive odors even if they do not contain glucose.

## 1. Introduction

The ability to memorize associations between different items and events enhances and modifies various behaviors, such as foraging, mating, and avoidance of stimuli that represent danger. The mechanisms of learning have been described in vertebrates and invertebrates, ranging from simple habituation to responses to novel stimuli [1,2,3,4,5,6]. Both holometabolous insects, such as honeybees [7,8,9], Diptera [10,11,12,13,14], and Lepidoptera [15,16], and hemimetabolous insects, such as Orthoptera [17,18] and cockroaches [19,20,21,22,23,24,25,26,27], have been investigated to understand associative learning using classical and operant conditioning. These insects show high performance in associating color, odor, and spatial information with rewards such as food rewards, and punishments such as electric shock and deterrents. Additionally, the learning abilities of disease vectors, including mosquitoes and kissing bugs [28,29,30] and pests such as cockroaches [31], have been of interest in pest management, because learning abilities modify the pattern of contact between insects and various resources (hosts and foods) or aversive materials including insecticides and entomopathogens. Understanding how learning ability modifies behaviors of disease vectors and pests will support the design and development of innovative pest control strategies.

The German cockroach, *Blattella germanica*, is a worldwide indoor pest not only causing sanitation problems via aggregation and fecal contamination, but also producing large amounts of allergens in their feces and other secretions [32]. Due to its omnivorous feeding habits, toxic baits, composed of phagostimulants and insecticides, have been the most effective tools against *B. germanica* since the mid-1980s. However, around 1990, researchers and pest control operators pointed out the emergence of behavioral resistance to toxic baits [33,34,35,36,37]. The most common form of behavioral resistance is glucose-aversion (GA), where cockroaches avoid eating this nutrient sugar in toxic baits [33,34,35,36,37]. The mechanism underlying GA is that cockroaches with this gustatory polymorphism detect glucose as a deterrent rather than a phagostimulant through gustatory receptor neurons (GRNs) housed in mouthpart sensilla [35,36]. Glucose aversion is heritable [38], and this trait can drive population replacement of wild type (WT) by GA cockroaches when glucose-containing toxic baits are used in cockroach control [37]. Our recent research has extended the GA phenotype to all glucose-containing sugars. Cockroach salivary enzymes degrade oligosaccharides, releasing glucose and thus extending GA from glucose to all glucose-containing oligosaccharides [39]. These findings also explain previous observations of aversions to disaccharides that contain glucose [40,41,42,43].

Food preferences and foraging strategy can be shaped by olfactory learning that associates odorants with tastants. While it is well known that German cockroaches are attracted to various food odors [44,45,46,47,48,49], few studies have addressed olfactory learning. In their foraging forays between the shelter and food sources, German cockroaches use food odors and landmarks, such as light sources [50,51,52,53]. Based on previous studies with the American cockroach (*Periplaneta americana*) using classical and operant conditioning [20,21,22,26,27], Liu and Sakuma (2013) [53] demonstrated that German cockroaches trained by associating menthol, an innate repellent, with sucrose, an innate phagostimulant, were attracted to menthol when presented alone. Conversely, cockroaches trained with vanillin as an innate attractant with sodium chloride as an innate deterrent, decreased their odor preference for vanillin.

The WT and GA cockroach strains have different feeding responses to sugars. While glucose and fructose are phagostimulants and caffeine is a deterrent for WT cockroaches, in GA cockroaches, fructose is a phagostimulant, and caffeine and glucose act as deterrents [35,36]. In this study, we hypothesized that the genetic differences of the gustatory perception systems of WT and GA cockroaches mediate different odor preferences by olfactory learning; namely, food odors might be associated with glucose as reward and as punishment for GA cockroaches. We trained groups of males using various innate attractants and three tastants (glucose, fructose, and caffeine), and then tested the modified odor preferences of individuals in two-choice olfactometers.

## 2. Materials and Methods

### 2.1. Insects

The wild type (WT) *B. germanica* colony (Orlando Normal) was collected in Florida in 1947 and has served as a standard insecticide-susceptible strain in many studies. The glucose-averse (GA) colony (T-164) was collected in Florida in 1989 and shown to be averse to glucose; continued artificial selection with a glucose-containing toxic bait fixed the homozygous GA allele(s) in this population (approximately 150 generations as of 2020) resulting in maximal sensitivity to glucose as a deterrent. All cockroaches were maintained on a rodent diet (Purina 5001, PMI Nutrition International, St. Louis, MO, USA) and distilled water at 27 °C, ~40% RH and a 12:12 h L:D cycle (8 p.m. to 8 a.m. photophase, 8 a.m. to 8 p.m. scotophase). We tested 10–18 days-old males during scotophase in this study. Since the feeding and foraging motivation of females are impacted by physiological changes related to the gonotrophic cycle, we tested only adult males.

### 2.2. Olfactometers

Each linear two-choice olfactometer was composed of two connected tubes (Figure 1A). The first was an acrylic acclimation chamber with a swivel metal screen gate. The upwind end of the acclimation chamber (near the gate) was connected to an acrylic bioassay tube with an acrylic divider sealed vertically in the upwind end. Each olfactometer was connected to a vacuum pump that provided a linear air velocity of 25 cm/s through the tube; the air was exhausted outside the building. Fluorescent lights covered with red photographic safety filters placed 60 cm below and above the olfactometers facilitated observation in the dark. In each bioassay, an individual cockroach was introduced into the acclimation chamber and allowed to acclimate to the airflow for 5 min, and then the swivel gate was gently opened. A single quiescent cockroach was activated by an odor; it walked upwind and made a choice of the right or left sides of the divided tube.

### 2.3. Lures Containing Odors

Each red rubber septum lure (#224100-020, Wheaton, Millville, NJ, USA) was placed in a 1.5 mL microcentrifuge tube. Undiluted odor compounds or compounds dissolved in solvents were applied to each lure (total volume 100 µL) and placed in the end of the divided bioassay tube. Following traditional associative learning terminology, we termed the training odor associated with a tastant during the conditioning session the conditioned stimulus (CS) (Figure 1B and Table 1).

### 2.4. Feeding Tubes Containing Tastants 

An aqueous tastant solution (total liquid volume 1.5 mL) was loaded into a feeding tube (1.5 mL microcentrifuge tube) and plugged with a small cotton ball. It was placed downwind of the lure of the divided bioassay tube, so that cockroaches could be exposed to specific odors as they fed. However, to prevent cockroaches from contacting the lure, a metal screen was placed between the feeding tube and lure. In the conditioning session, cockroaches were allowed to freely drink each of the two tastant solutions. Based on our previous findings [35,36], we used the 90% effective concentration (EC_90_) for each tastant, obtained from dose-feeding response curves: 300 mM fructose (EC_90_ for appetitive responses in both strains), 1000 mM glucose (EC_90_ for appetitive responses in WT and aversive responses in GA cockroaches), and 10 mM caffeine (EC_90_ for aversive responses in both strains). The taste quality (valence) of glucose and fructose is processed by the same phagostimulant-GRNs mediating appetitive feeding responses in both WT and GA cockroaches, and the taste quality of caffeine is processed by the same deterrent-GRNs mediating aversive responses in both strains [36,37]. In GA cockroaches, however, glucose is also processed by the same deterrent-GRNs that process caffeine and mediate aversive responses. Studies with honeybees have shown that consumption of and contact with caffeine influences memory formation [54,55]. However, since 90% of the cockroaches avoid 10 mM caffeine immediately after contacting it with their mouthparts, and they do not consume it [35,36], we expected that caffeine would serve as a deterrent that stimulates deterrent-GRNs but not the central nervous system (CNS). In WT cockroaches, fructose and glucose were rewards or appetitive unconditioned stimuli (US+) for a positive association with odors, and caffeine was a punishment or aversive unconditioned stimulus (US−) for a negative association with odors (Figure 1B and Table 1). In GA cockroaches, fructose acted as a reward, whereas caffeine and glucose were deterrent and served as punishment. When cockroaches associated a CS with US+ by conditioning (training), they were expected to prefer the CS in preference assays. When they associated CS with US−, they were expected to avoid or ignore the CS. 

### 2.5. Chemicals

All chemicals for odors, tastants, and solvents (mineral oil and ethanol) were obtained from Sigma Aldrich (St. Louis, MO, USA), except vanilla and chocolate extracts. Odors: beta-caryophyllene (purity ≥80%), farnesene (≥99%), benzaldehyde (≥99%), 3-methyl-1,2-cyclopentanedione (≥99%), 4-hydroxy-3-methoxybenzaldehyde (vanillin) (≥99%), 4,5-dimethyl-2-ethylthiazole (97%), and 2,4,5-trimethylthiazole (98%). Vanilla (All Natural Pure Vanilla Extract, McCormick, Hunt Valley, MD, USA) contains vanilla bean extractives in water and alcohol. Chocolate (Chocolate Extract, OliveNation, Avon, MA, USA) contains water, alcohol, chocolate concentrate and vanilla extract. These two odors were obtained from local grocery stores. Tastants: glucose (99.9%), fructose (≥99%) and caffeine (99%).

### 2.6. Conditioning

The divided bioassay tube formed a linear two-choice assay tube (Figure 1A), and was used for conditioning, defined as the training of insects before bioassays. One day before any bioassays or conditioning, 20 adult males (10–12 days-old) were placed in a plastic cage (T-79, Althor Products, Windsor Locks, CT, USA) containing distilled water, a rodent diet, and an egg carton shelter. The cage had two ports in line with each other to accept the bioassay tube upwind (air inlet) and a downwind exhaust tube (Figure 1B and Table 1). These ports were kept closed during the 1 day acclimation phase. In the unconditioned odor preference assay, individual males were obtained directly from this cage. In the conditioning session for the other three bioassays, the downwind port was connected to a vacuum pump, as described in Section 2.2 (Olfactometer). The upwind port was connected to the bioassay tube. After removing the rodent diet from the arena, feeding tubes and lures were placed upwind in the divided portion of the bioassay tube. Cockroaches were allowed to freely visit the feeding tubes and thus trained themselves to associate tastants and odors for 1 h from 12 to 1 pm, at nighttime of the reversed L–D cycle, when cockroaches actively forage [56]. All insects were attracted to odor sources and contacted the feeding tubes within 5 min after starting the conditioning, and all of them returned to the cage (and shelter) within 1 h. After the conditioning session, the ports were closed, the rodent diet was returned to the cage, and males were kept in the cage until the next conditioning session or bioassays (Figure 1B). 

Traditional classical and operant conditioning rely on the assumption that changes in conditioned response probability observed during training adequately represent neuronal plasticity, and commonly, behavioral plasticity is quantified by averaging over a population of identically treated animals [1,2]. On the other hand, recent studies using honeybees [57] and the American cockroach [27] suggest that even if insects were trained by well controlled methods, the average behavioral score of the group does not represent individual behavior, which is driven by unique personality traits. The individual learning of such species may be influenced by various factors including their sensory sensitivity, their ability to learn a task, the speed of learning, and their asymptotic performance. In this study, however, we did not identify individuals, because the group of insects was self-trained. Therefore, learning curves of individuals were not evaluated during the conditioning session, and, in Bioassays 2–4, we tested the retained olfactory memory as the average behavioral score of groups after training.

### 2.7. Bioassay Procedures

Bioassays were conducted between noon and 4 p.m. (Table 1). Unless stated otherwise, a single male was tested only once in a clean olfactometer. When the male was quiescent in the acclimation chamber, the gate was opened carefully, the lures containing odor stimuli were introduced at the upwind end of the olfactometer, and the insect’s response was noted by direct observation. A positive response was scored when the male entered the divided bioassay tube within 2 min and remained there for 15 s. After each bioassay, each olfactometer was flushed out with fresh air for 2 min. The positions of the two lures were randomly switched between the left and right sides of the divided section of the bioassay tube. Every five bioassays, the olfactometers were washed with distilled water and ethanol. 

### 2.8. Bioassay 1: Unconditioned Odor Preference

Considering that, in their natural environment, cockroaches approach certain innately preferred odors and avoid innately repellent odors, we screened for attractive food odors in various food sources, including plant materials and human food for use in Bioassays 2 through 4 (Table 1). Distilled water, ethanol, and mineral oil were used as solvents. As general odors contained in sweet snacks and chocolate drinks, we used 3-methyl-1,2-cyclopentanedione (coffee and caramel flavor, 10^−4^, 10^−3^, 0.01, 0.1, 1, 10 µg in 100 µL mineral oil per lure), 4,5-dimethyl-2-ethylthiazole (burnt hazelnut odor, 10^−8^, 10^−7^, 10^−6^, 10^−5^, 10^−4^, 10^−3^ µg in 100 µL mineral oil per lure) and 2,4,5-trimethylthiazole (musty, nutty, and brown cocoa and coffee odor, 10^−6^, 10^−5^, 10^−4^, 10^−3^, 0.01, 0.1 µg in 100 µL mineral oil per lure). As general plant terpenes, we used beta-caryophyllene (sweet woody spicy and peppery odor) and farnesene (sweet, woody, herbal, and green aroma) at 10^−4^, 10^−3^, 0.01, 0.1, 1, and 10 µg in 100 µL mineral oil in each lure. As aromatic aldehydes, we used benzaldehyde (almond odor, 10^−4^, 10^−3^, 0.01, 0.1, 1, 10 µg in 100 µL mineral oil per lure) and vanillin (vanilla odor, 4-hydroxy-3-methoxybenzaldehyde, 10^−4^, 10^−3^, 0.01, 0.1, 1, 10 µg in 100 µL ethanol per lure). As a blend of sweet odors, commercial vanilla extract (All Natural Pure Vanilla Extract, McCormick, Hunt Valley, MD, USA) and chocolate extract (OliveNation, Avon, MA, USA) were dissolved in distilled water at 0.01, 0.1, 1 equivalents of the original product to find the optimal concentrations for our bioassays. Each odor was applied to a single red rubber septum lure and placed in one side of the divided bioassay tube. A solvent-only lure was placed in the other side of the divided tube. We tested 20–30 insects at each concentration of each odor source. Additionally, a two-choice test using vanilla and chocolate was conducted with the undiluted original products to assess the unconditioned (innate) odor preferences for these two attractive odors (Appendix A).

### 2.9. Bioassay 2: Conditioned Odor Preference after Conditioning with a Single Odor

To test whether the insects associated odor with either rewarding or punishing tastants, cockroaches self-trained (operantly conditioned) with a combination of a single odor and a single tastant in the conditioning session; then, olfactometer bioassays were carried out (Table 1). During conditioning, one side of the divided bioassay tube contained a single combination of ‘odor (lure) + tastant (feeding tube)’. Six types of combinations were prepared: ‘Vanilla (CS) + Frucotse, Glucose or Caffeine (US+ or US−)’ and ‘Chocolate (CS) + Fructose, Glucose or Caffeine (US+ or US−)’. The other side of the tube contained a lure and the feeding tube contained distilled water. To test the impact of the training, we tested two types of conditioning. The first was a single conditioning session of 1 h, after which the insects were tested 24 h later for their odor preference in the two-choice preference assay using both vanilla and chocolate odors (Table 1). The second training paradigm was three successive 1 h conditioning sessions (1 h at 12–1 p.m. each day for three days), followed by odor preference assays approximately 24 h later. In a comparison of odor preference among the treatments, we used the results of the innate odor preferences from Bioassay 1 as control. If trained cockroaches chose the CS more than untrained cockroaches did, we considered that they associated the CS with the US+. If trained cockroaches preferred the CS less than the untrained cockroaches did, we considered that they associated CS with the US−. We tested 30–40 males in each treatment.

### 2.10. Bioassay 3: Conditioned Odor Preference after Conditioning with Two Odors 

To test whether insects associated two combinations of tastants and odors, males were trained with both vanilla and chocolate odors using two types of tastants. The divided tubes contained different combinations of ‘lure + tastant’: ‘Either Vanilla or Chocolate + either Fructose or Caffeine’, ‘Either Vanilla or Chocolate + either Fructose or Glucose’, and ‘Either Vanilla or Chocolate + either Glucose or Caffeine’. Males received either one or three successive 1 h conditioning sessions. In this paradigm, both vanilla and chocolate used in the two-choice olfactometer bioassays acted as CS associated with either US+ or US− (Table 1). Data analysis was by the same methods described in Section 2.9 (Bioassay 2). 

### 2.11. Bioassay 4: Retention of Olfactory Memory

To test the retention of olfactory memory, we exposed insects to three successive 1 h conditioning sessions (1 h at 12–1 p.m. each day for three days). The combinations of ‘lure (CS) + tastant (US)’ were ‘Vanilla + Fructose and Chocolate + Caffeine’, ‘Vanilla + Fructose and Chocolate + Glucose’, and ‘Vanilla + Glucose and Chocolate + Caffeine’. Conditioned preference bioassays with vanilla and chocolate were conducted 2, 3, or 5 days later (Table 1). Data analysis was, by the same methods, described in Section 2.9 (Bioassay 2).

### 2.12. Data Analysis

The percentage of males responding was calculated by the formula: % responding = # of insects making a choice/total # of tested insects. Significant differences across treatments and strains were compared using Chi-square tests with Bonferroni corrections (α = 0.05/# of comparisons). The percentage choice (preference) for each lure was calculated as: % choice = # of insects choosing the lure either on the right side or left side / total # of insects making a choice. Preference between lures was tested by Chi-square test (α = 0.05). Significant differences among the treatments in each strain were tested with Chi-square tests with Bonferroni corrections. R-4.1.0 with js-STAR as an open-source tool for working on R programs was used for the statistical analysis.

## 3. Results

### 3.1. Innate Odor Preferences 

Despite their different genetic backgrounds, WT and GA males responded similarly to the three solvents and the seven odorants we tested. For the three types of solvents, only 10% of tested insects walked upwind in the bioassay olfactometer, and they equally chose the left and right sides of the divided tube (Figure 2A and Appendix A), confirming no side-bias in the olfactometers. The % responding to each of six different odorants significantly increased in a dose-dependent manner (Figure 2B–G). However, the % responding to benzaldehyde decreased at high concentrations (Figure 2H), indicating that this odor acted as a repellent at high concentration, as shown in other insects including *Drosophila* [58]. In the comparison of insects making a choice in each strain, the % choice, a measure of discrimination between the odor and solvent, increased in a dose-dependent manner for four compounds (Figure 2B,C,E,G), but three compounds (2,4,5-trimethylthiazole, farnesene and benzaldehyde) stimulated insects to respond, but they could not discriminate the odors from the solvent control at any of the concentrations (Figure 2D).

On the other hand, the commercial flavor extracts of vanilla and chocolate attracted cockroaches of both strains more than any of the single compounds (Figure 3A, Appendix A). Over 80% of the insects responded in the olfactometers at all tested concentrations. Within 60 sec they made clear choices for the undiluted odors. When they were given a choice between vanilla and chocolate, 90% of the males approached the odor sources and significantly more insects chose chocolate over vanilla (Figure 3B, Appendix A). Importantly, WT and GA males responded similarly to vanilla and chocolate and to the dual-choice of both odors. Therefore, we used undiluted concentrations (1 equivalent) of commercial vanilla and chocolate extracts in Bioassays 2–4.

### 3.2. Association of a Single Odor with Tastants

Figure 4 shows that the odor preferences of cockroaches could be modified after six types of conditioning. In a comparison of the number of insects choosing odors across various treatments (% choice), the trained insects significantly chose the CS odor associated with US+ (Figure 4A, fructose for the two strains; Figure 4B glucose for WT). No significant differences were found between the single conditioning session and the three successive conditioning sessions, suggesting that a single h of conditioning was sufficient to modify the odor preference with a reward and that multiple conditioning sessions did not significantly change the odor preference modified by a single conditioning session (Appendix A). 

On the other hand, after conditioning with US− in association with either vanilla or chocolate, insects that experienced three conditioning sessions did not choose the CS odors over the unconditioned odors (Figure 4B, glucose for GA cockroaches; Figure 4C caffeine for the two strains). We found no significant differences in the odor preferences between control insects and insects exposed to one 1 h conditioning session. These results indicate that (a) the innate preferences for vanilla and chocolate were modified by their olfactory association with tastants, (b) the insects needed more training to associate odors with punishment than with reward, and (c) that while WT cockroaches associated glucose with odors in appetitive rewarding foraging contexts, GA cockroaches associated glucose with odors as punishment and avoided both. No significant differences were found in the % of insects responding among treatments and strains (Chi-square tests with Bonferroni corrections, *p* > 0.0003), except in the way GA cockroaches perceived glucose as a deterrent and therefore avoided odors associated with it (Figure 4B).

### 3.3. Association of Two Odors with Tastants 

The modified odor preferences after conditioning with two combinations of odors and tastants are shown in Figure 5. In a comparison of the number of insects making a choice (% choice), both strains trained with the ‘Vanilla + Fructose’ and ‘Chocolate + Caffeine’ combination preferred vanilla significantly more than the unconditioned control insects did (Figure 5A left). In the opposite conditioning, both strains chose chocolate significantly more than the unconditioned insects did (Figure 5A right). Although there were no differences in the modified odor preferences for rewards between one and three conditioning sessions in the experiment using single odors (Figure 4A for WT and GA and 4B for WT), when using two odors, three conditioning sessions were significantly more effective in modifying odor preferences than one conditioning session (Figure 5A). 

In the conditioning paradigm with either combinations of ‘Vanilla + Fructose’ and ‘Chocolate + Glucose’ or ‘Vanilla + Glucose’ and ‘Chocolate + Fructose’, both fructose and glucose acted as rewards (US+) for WT cockroaches. They tended to lose preference for chocolate over vanilla, but with no significant difference from the control (Figure 5B WT). On the other hand, fructose and glucose acted as reward (US+) and punishment (US−), respectively, for GA cockroaches. They significantly preferred odors associated with fructose but not with glucose (Figure 5B GA). 

Figure 5C shows the odor preferences after conditioning using either combinations of ‘Vanilla + Glucose’ and ‘Chocolate + Caffeine’ or ‘Vanilla + Caffeine’ and ‘Chocolate + Glucose’. For WT cockroaches, glucose and caffeine acted as US+ and US−, respectively, but both acted as US− for GA cockroaches. WT males preferred odors associated with glucose (Figure 4C WT). However, the odor discrimination (% choice) of GA males was not modified by conditioning. Instead, the % responding significantly decreased from 90% in the controls to 57.1–65.7% following one conditioning session, and 37.1–42.5% after three conditioning sessions (Chi-square test with Bonferroni corrections, *p* < 0.0003) (Figure 5C GA). The percentage responding in all other treatments was >80% with no significant differences among the treatments. The results suggest that GA cockroaches associated both vanilla and chocolate odors with punishment due to distasteful caffeine and glucose and stopped approaching otherwise attractive odor sources.

### 3.4. Retention of Olfactory Memory

Retention of olfactory memory was compared among the controls (innate unconditioned preferences) and 1, 2, 3, and 5 days after operant conditioning. After three conditioning sessions using ‘Vanilla + Fructose and Chocolate + Caffeine’, the modified odor preference was highest 1 day later, and it gradually returned to the level of the control within 5 days (Figure 6A). In the conditioning with ‘Vanilla + Fructose and Chocolate + Glucose’, WT males quickly lost their innate odor preference and retained their modified equal preference of the two odors (vanilla and chocolate) for at least 3 days; there were no significant differences among the four post-conditioning preferences and the innate preferences (Figure 6B). GA cockroaches showed a similar pattern of percentage choice as in Figure 6A, since both caffeine (Figure 6A) and glucose (Figure 6B) were distasteful and acted as punishment (US−). After conditioning with ‘Vanilla + Glucose and Chocolate + Caffeine’ (Figure 6C), WT males showed a similar pattern as in Figure 6A—they avoided the innately preferred chocolate odor because it was paired with caffeine during conditioning but regained the chocolate preference within 3 days. There were no significant differences in odor preferences among treatments in GA males, but the percentage responding decreased significantly one and two days after conditioning and recovered 3 days later. The results suggest that aversive olfactory memory for punishments is retained in GA cockroaches for 3 days.

## 4. Discussion

German cockroaches live in anthropogenic environments where food is available, such as kitchens, food-handling areas (e.g., in restaurants, hospitals, airplanes), pet shops, and farm structures where animals are housed. They usually make nighttime foraging trips from aggregation sites in search of food. Their odor-guided foraging behavior and olfactory learning abilities are important for survival in various environments. If the odors and tastes associated with food sources are appetitive, and their presence is persistent and predictable, cockroaches visit them repeatedly, guided by learning the location of landmarks, odors and light sources [50,51]. In this study, using two cockroach strains that differ in the way they perceive glucose, we demonstrated the importance of taste in modifying preferences of odors associated with food.

### 4.1. Innate Odor Preferences

The WT and GA cockroach strains showed similar innate odor preferences (Bioassay 1). Cockroaches were attracted to single odorants in a dose-dependent manner (except to benzaldehyde), but their discrimination between the odor lures and the solvent control lure was significantly weaker than their preferences for commercial extracts of vanilla and chocolate. Considering that the commercial extracts contain multiple compounds from natural sources, these findings suggest that blended odors may enable greater resolution and discrimination of odors. Our findings on the innate (without training) behavioral responses to individual odorants are consistent with previous work. Of 30 substances screened for attractiveness, only one was significantly attractive to the German cockroach [45]. This is not surprising, because food items are never represented by pure odorants and cockroaches encounter complex odors from human and animal foods. The German cockroach is attracted to a variety of complex food odors including peanut butter, banana, bread, beer, distiller’s grain, and pet foods [49]. Nevertheless, we cannot dismiss the idea that the olfactometer conditions were not optimized for the individual odor compounds, for example, by manipulating the flow rate and release rate of each odorant. It is also plausible, though unlikely, that differences in olfactory sensitivity of individual cockroaches for each compound resulted in mediocre group averages. However, there are no reports on differences in odor sensitivity profiles of olfactory receptor neurons (ORNs) for various single compounds among individuals or strains.

### 4.2. Modification of Odor Preferences after Self-Training

Conditioning parameters to test long-term memory in *P. americana* are well designed and have been the subject of extensive studies [20,21,26,27]. Liu and Sakuma [53] employed one of the approaches of operant conditioning. They also evaluated the olfactory memory of German cockroaches by the number of visits they made to an odor source and confirmed their olfactory learning ability. In contrast, to create a natural foraging experience that cockroaches might encounter with baits, we allowed groups of cockroaches to self-train without us controlling their conditioning. Despite the less controlled conditioning, three tastants significantly modified the odor preferences of the cockroach groups after they were associated with vanilla or chocolate odors. That is, our results strongly support previous studies showing that odor preferences increased when associated with rewarding (appetitive) tastants and decreased when associated with punishing (aversive) tastants (Bioassay 2). It is known that sugars such as sucrose constitute strong rewards to form olfactory memory in insects, including in *P. americana* [20]. A single training event (one-time stimulation with peppermint paired with sucrose) was sufficient to alter *P. americana*’s odor preference, and the olfactory memory formed by three conditioning sessions was retained for at least 4 weeks. A previous study using the German cockroach [53] reported that insects trained by pairing menthol with sucrose increased their odor preference for menthol after three training events (2 s/training), and the olfactory memory was retained 1 day after training. Our study demonstrated that a single self-training session with a rewarding tastant also induced changes in odor preferences of grouped insects and the cockroaches retained the olfactory memory for 24 h. On the other hand, it is common to use deterrents such as sodium chloride, saline, and quinine as punishment in cockroaches [20,21] and honeybees [59] to test whether an aversive tastant modified odor preference. In our case, we tested caffeine as punishment to compare the impact of glucose sensory processing in olfactory memory formation, because caffeine stimulates the same deterrent-GRNs that process glucose as a deterrent in GA cockroaches [35,36]. Although caffeine is known as a CNS stimulant that can modify and enhance olfactory learning performance via ingestion and contact in honeybees [54,55], the 10 mM of caffeine we tested was enough for 90% of cockroaches to avoid ingesting it after contacting caffeine with their mouthparts. Liu and Sakuma [53] conducted aversive conditioning with *B. germanica* using saline punishment with either vanillin or menthol odors. They showed decreased odor preference 30 min after training, but it remained unknown if long-term memory (>24 h) could modify odor preferences in association with deterrents. Our result clearly showed that cockroaches retained the modified preference for odors that were associated with aversive tastants for at least 24 h.

An additional finding of this study was that different numbers of conditioning sessions were required to impose modifications of odor preference by rewarding and punishing tastants. The results of Bioassay 2 indicated that 1 h of operant conditioning using rewarding tastants was sufficient to increase their odor preference 24 h later. In contrast, training using tastants as punishment required three 1 h conditioning sessions (one each day) to modify the odor preferences (Bioassay 2). In the olfactory associative learning, environmental odors are received by ORNs housed in sensilla on peripheral chemosensory appendages (e.g., antennae) and chemical information is translated to neural signals. The neural signals from the ORNs are processed in primary olfactory centers (antennal lobe) and secondary olfactory centers (mushroom body and lateral horn) in the CNS. The modulation of olfactory behaviors based on non-associative experiences, such as sensitization and habituation, and associative experience, is mediated by structural and physiological changes in this pathway. Much progress has been made on the mechanisms underlying these forms of learning, and it has been demonstrated that the secondary olfactory center in the brain is an important region for integration of information from different sensory modalities such as vision and gustation. Indeed, the majority of structural and physiological modifications due to associative long-term memory, which is formed via functional protein synthesis, are observed at the CNS level [6,12,60,61]. Under well controlled classical and operant conditioning, the experimenter and not the insects choose the frequency and exposure periods for paired and unpaired association of odors and tastants. In contrast, in our study, free-moving insects self-trained and could leave the deterrents (punishment) at will and contact and spend more time at the appetitive tastants (reward). During self-training, however, the number of visits and the time spent at each stimulus are mediated by the quality and concentration of tastants and interactions among individuals. Therefore, differences in these behaviors might influence the differential memory formation with rewards and punishments. In *Drosophila*, it is known that appetitive and aversive long-term memories are formed by different neural pathways via classical conditioning [12]. Our results suggest that the German cockroach may have different circuits for appetitive and aversive memories, like other insects, and that modifications of odor preferences with reward and punishment appear independent of each other.

We also found that the preference for an odor that was associated with a rewarding tastant was enhanced when the cockroaches formed two types of memories by receiving two pairs of odor + tastant combinations (odor + reward and odor + punishment, Bioassay 3). Our results suggest that olfactory memory and odor preferences are modified by both appetitive and aversive memories. After training by two pairs of ‘odor + reward’, the insects tended to prefer the two odors nearly equally in foraging behavior. After training by two pairs of ‘odor + punishment’, the insects did not approach both odor sources. These findings indicate that cockroaches can associate multiple odors with either reward or punishment at the same time. Such memory was retained for at least three days (Bioassay 4).

### 4.3. Olfactory Learning and Behavioral Resistance in Cockroaches

Previous studies demonstrated the function of glucose as a deterrent in feeding and foraging, and it had been suggested that other sugars such as fructose and sucrose could serve as alternative phagostimulants for newly designed toxic baits for controlling GA cockroach populations [33,35,36,37]. However, recently, we found that most candidate oligosaccharides, such as sucrose and maltose, rapidly release glucose as salivary enzymes degrade complex sugars during feeding, resulting in interrupted consumption of test solutions in GA cockroaches [39]. These observations suggest that the GA phenotype is substantially extended to all oligosaccharides that contain glucose. As these alternative sugars are applied as phagostimulants to newly formulated toxic baits, GA cockroaches will refrain from satiating on these baits, resulting in control failures. 

Perhaps our most significant finding was that glucose acted as punishment in olfactory learning for GA cockroaches, while it acted as reward for WT cockroaches. Thus, our results indicate that the effect of glucose as aversive tastant impacts not only simple feeding rejection but also odor-guided foraging strategy. Generally, toxic baits are placed for several months as part of the pest control procedure. WT cockroaches that are attracted to the baits are then eradicated by consuming the active ingredients. However, GA cockroaches that are attracted to the baits reject eating them as they experience the baits’ aversive taste. This repeated unsatisfying foraging experience may contribute to the formation of an aversive olfactory memory. Even if the newly formulated baits do not contain aversive tastants, there is the possibility that GA cockroaches might avoid such baits based on their aversive olfactory memory. This foraging strategy would allow cockroaches to find alternative, safer food sources in the human environment, and after finding these favorite food sources, they will form an appetitive olfactory memory of them. Thus, the efficacy of the new baits might be compromised. Therefore, olfactory learning and memory, together with salivary enzymes, effectively extend the GA phenotype beyond the basic mechanisms of altered glucose interaction with gustatory receptors. 

The development of baits and their proper deployment are highly dependent on comprehensive understanding of cockroach behavior mediated by various sensory capabilities, including olfactory, gustatory, visual, and mechanosensory performance. The consideration of population dynamics reflecting changes in demography, insecticide- and behavioral-resistance patterns, and food preferences based on nutrient requirement is also important in the design of pest control programs [56,62]. Our findings highlight the need to consider olfactory learning to improve control not only of cockroaches, but also other pest insects that evolved behavioral resistance.

## Figures and Tables

**Figure 1 insects-12-00724-f001:**
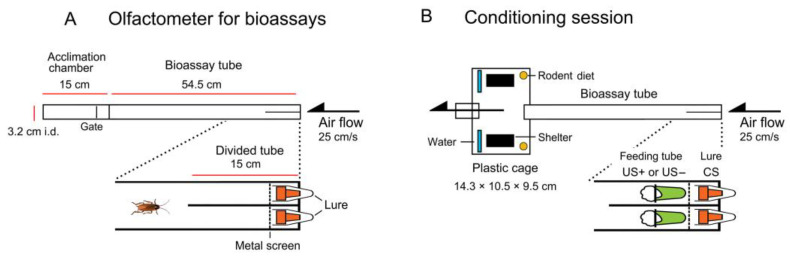
Olfactometer schematics. (**A**) A linear two-choice olfactometer was composed of an acclimation chamber and bioassay tube with two odor sources. (**B**) A cage containing a group of male cockroaches was connected to the bioassay tube. A feeding tube containing a tastant was paired with a lure containing an odor source in one side of the divided bioassay tube and another pairing of tastant and odor lure was placed in the other side of the tube.

**Figure 2 insects-12-00724-f002:**
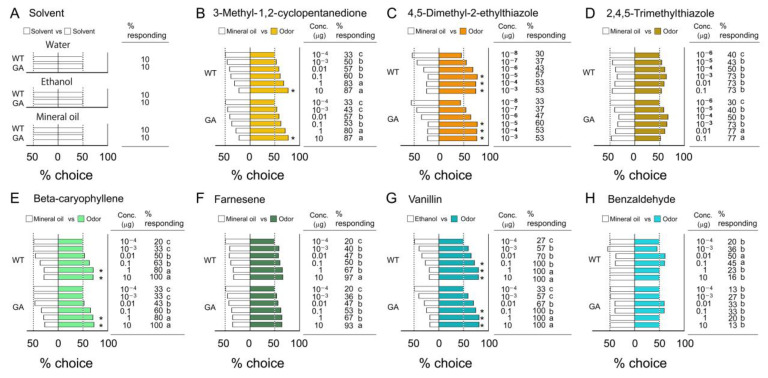
Responses to solvents and innate preferences for single odorants (Bioassay 1). Innate preferences (% choice) and the % of insects that made a choice (% responding). (**A**) Effects of three solvents on the discrimination ability of males between the left and right sides of the divided tubes. *N* = 20 for each concentration-solvent combination. (**B**–**H**) Odor preferences for seven compounds. Total number of tested insects was 20–30 for each treatment. An asterisk (*) denotes a significant preference for the odor over the solvent control (Chi-square test, *p* < 0.05). Different letters indicate significant differences in % responding to each compound across concentrations and strains (for (**B**–**H**), Chi-square tests with Bonferroni corrections, *p* < 0.0007 (0.05/66)) (Appendix A).

**Figure 3 insects-12-00724-f003:**
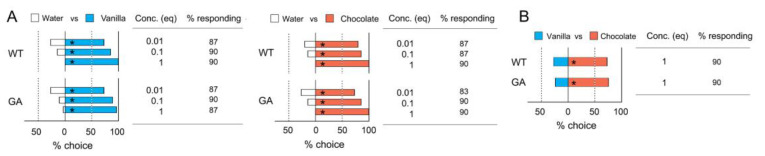
Innate odor preferences for blends of multiple compounds (Bioassay 1). (**A**) Innate preferences for commercial extracts of vanilla and chocolate vs. water in WT and GA cockroaches. No significant differences were found in the % responding among the concentrations and between cockroach strains (Chi-square tests with Bonferroni corrections, *p* > 0.0033 (0.05/15)). (**B**) % of insects choosing vanilla vs. chocolate in two-choice assays. WT and GA cockroaches significantly preferred chocolate over vanilla. An asterisk (*) in both (**A**,**B**) denotes a significant preference for the side with greater % choice (Chi-square test, *p* < 0.05). Concentration is presented in terms of extract equivalents with 1 representing the undiluted extract. The total number of tested insects was 30–40 for each experiment (Appendix A).

**Figure 4 insects-12-00724-f004:**
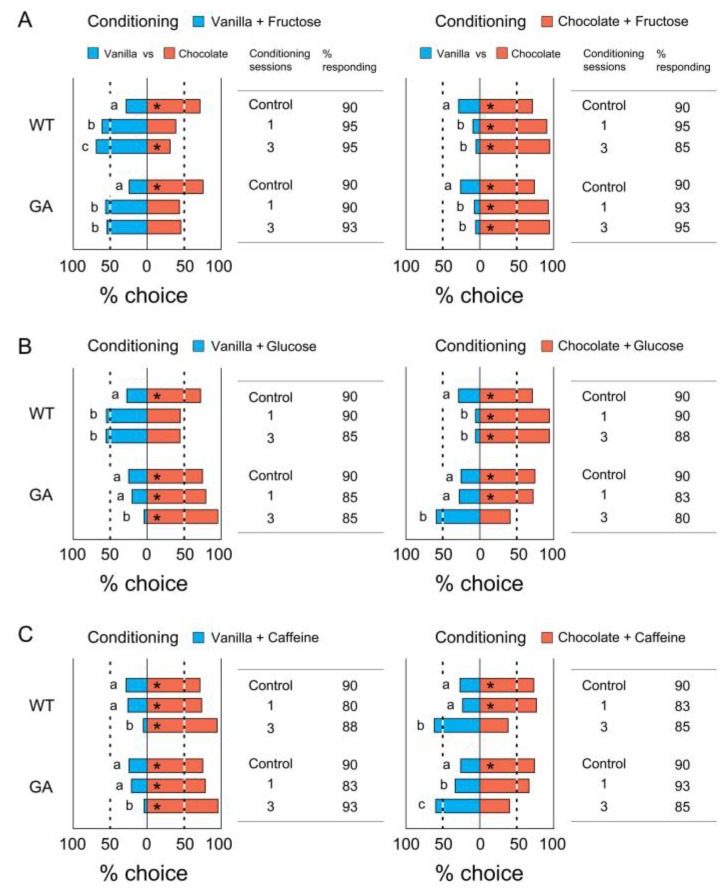
Modified odor preferences for vanilla and chocolate in WT and GA cockroaches after one or three odor conditioning sessions to associate an odor (Vanilla or Chocolate) with reward (US+) or punishment (US−) (Bioassay 2). The % choice of control males shown in all figures were obtained from the innate unconditioned odor preferences (Figure 3B, Appendix A). (**A**) % choice and % responding for the combination of ‘either Vanilla or Chocolate + Fructose (US+)’; (**B**) for the combination of ‘either Vanilla or Chocolate + Glucose (US+)’ in WT cockroaches, or the combination of ‘either Vanilla or Chocolate + Glucose (US−)’ in GA cockroaches; and (**C**) for the combination of ‘either Vanilla or Chocolate + Caffeine (US−)’. An asterisk (*) denotes a significant preference for the side with greater % choice (Chi-square test, *p* < 0.05). Different letters associated with % choice indicate significant differences in the number of insects choosing odors among treatments within each strain (Chi-square tests with Bonferroni corrections, *p* < 0.0167 (0.05/3)). There were no significant differences in the % insects responding among the treatments and between cockroach strains (Chi-square tests with Bonferroni corrections, *p* > 0.0003 (0.05/15)). The total number of tested insects was 40 for each treatment (Appendix A).

**Figure 5 insects-12-00724-f005:**
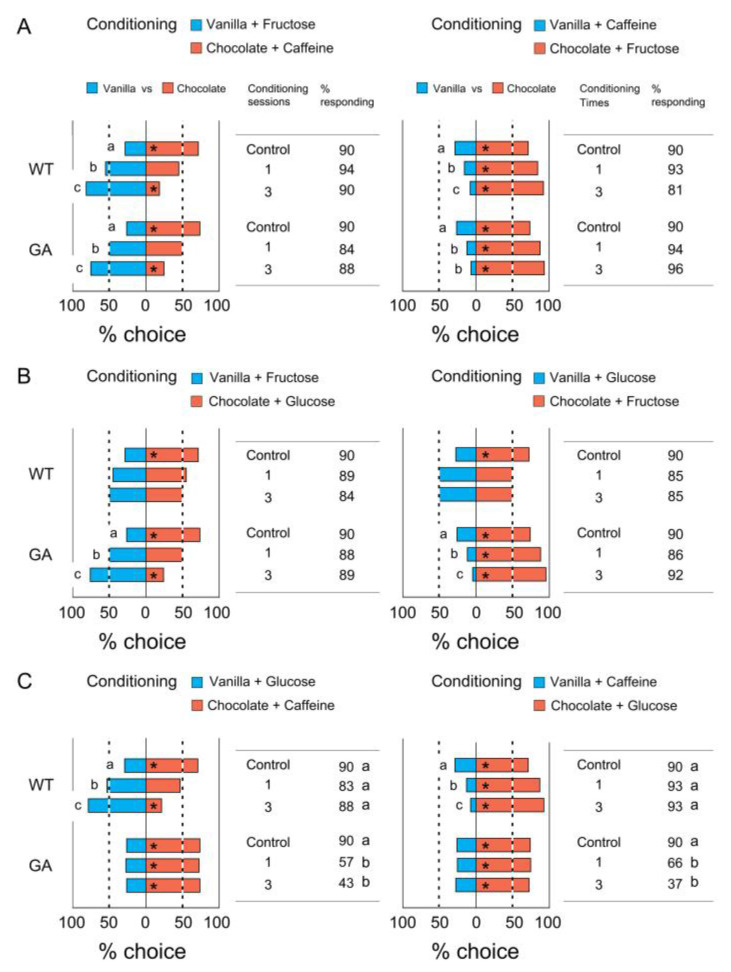
Conditioned odor preferences after association of two odors with reward or punishment. Modified odor preferences for vanilla and chocolate in WT and GA cockroaches after conditioning using two combinations of odors and tastants (Bioassay 3). The % choice of control shown in all figures were obtained from the innate unconditioned odor preferences (Figure 3B, Appendix A). (**A**) % choice and % responding for the combinations of ‘Vanilla + Fructose (US+) and Chocolate + Caffeine (US−)’ and ‘Vanilla + Caffeine and Chocolate + Fructose’, showing that both WT and GA males avoided odors associated with caffeine. (**B**) % choice and % responding for the combinations of ‘Vanilla + Fructose and Chocolate + Glucose’ and ‘Vanilla + Glucose and Chocolate + Fructose’, showing that both fructose and glucose acted as US+ for WT males, but glucose acted as US− for GA males. (**C**) % choice and % responding for the combination of either ‘Vanilla + Glucose and Chocolate + Caffeine’ or ‘Vanilla + Caffeine and Chocolate + Glucose’, showing that GA males expressed their innate odor preference when both odors were paired with punishment (caffeine or glucose). An asterisk (*) denotes a significant preference for the side with greater % choice (Chi-square test, *p* < 0.05). Different letters associated with % choice indicate significant differences in the number of insects choosing odors among treatments (Chi-square test with Bonferroni corrections, *p* < 0.0167 (0.05/3)). Different letters associated with % responding indicate significant differences in the % insects responding among the treatments and between cockroach strains (Chi-square test with Bonferroni corrections, *p* > 0.0003 (0.05/15)). The total number of tested insects was 40–105 for each treatment (Appendix A).

**Figure 6 insects-12-00724-f006:**
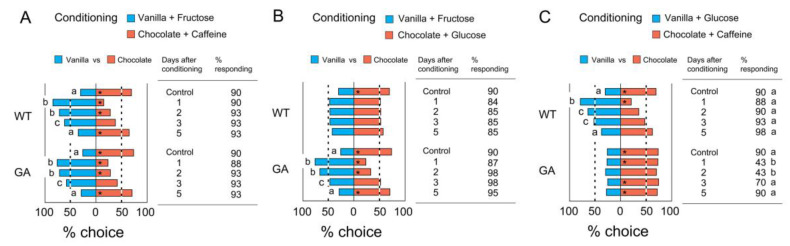
Retention of olfactory memory (Bioassay 4). Odor preferences for vanilla and chocolate after three successive 1 h conditioning sessions with two CSs were each associated with different US. Controls, unconditioned innate preferences) and 1 day later (1) were obtained from Figure 5. (**A**) % choice and % responding after conditioning with combinations of ‘Vanilla + Fructose (US+) and Chocolate + Caffeine (US−)’, showing a return to a preference for chocolate within 5 days after conditioning. (**B**) % choice and % responding after conditioning with combinations of ‘Vanilla + Fructose and Chocolate + Glucose’, also showing in GA males the fading of memory within 5 days of the association of distasteful glucose with innately preferred chocolate. (**C**) % choice and % responding after conditioning with combinations of ‘Vanilla + Glucose and Chocolate + Caffeine’, showing again that, as memory faded, GA males expressed their innate odor preference for chocolate when both odors were paired with punishment (caffeine or glucose). An asterisk (*) denotes a significant preference for the side with greater % choice (Chi-square test, *p* < 0.05). Different letters associated with % choice indicate significant differences in the number of insects choosing odors among treatments (Chi-square test with Bonferroni corrections, *p* < 0.005 (0.05/10)). Different letters associated with % responding indicate significant differences in the % insects responding among the treatments and between cockroach strains (Chi-square test with Bonferroni corrections, *p* > 0.0011 (0.05/45)). The total number of tested insects was 40–50 for each treatment (Appendix A).

**Table 1 insects-12-00724-t001:** Symbols of conditioning and schedule of experiments.

Symbols of Conditioning Session in Figure 1B		
Symbols (Terms)	Compounds Used	
CS (Conditioning stimulus)	Odors of vanilla and/or chocolate	
US + (Appetitive unconditioned stimulus)	Reward tastant (fructose and glucose for WT, fructose for GA)	
US − (Aversive unconditioned stimulus)	Punishment tastant (caffeine for WT, caffeine and glucose for GA)	
**Schedule of experiments**		
	**Conditioning** (1 session = 1 h/day, 12–1 p.m. [scotophase])	**Two-choice Bioassay** (12–4 p.m. [scotophase])
Bioassay 1	No conditioning	Single odor vs. solvent Vanilla vs. chocolate
Bioassay 2	1 or 3 sessions One type of ‘CS + US’: vanilla + either fructose, glucose, or caffeine; chocolate + either fructose, glucose, or caffeine	24 h after conditioning Vanilla vs. chocolate
Bioassay 3	1 or 3 sessions Two types of ‘CS + US’: vanilla + either fructose or caffeine and chocolate + either fructose or caffeine; vanilla + either fructose or glucose and chocolate + either fructose or glucose; vanilla + either glucose or caffeine and chocolate + either glucose or caffeine.	24 h after conditioning Vanilla vs. chocolate
Bioassay 4	3 sessions Two types of ‘CS + US’: vanilla + fructose and chocolate + caffeine; vanilla + fructose and chocolate + glucose; vanilla + glucose and chocolate + caffeine.	2, 3, or 5 days after conditioning Vanilla vs. chocolate

## Data Availability

Data associated with this manuscript have been archived in Dryad Digital Repository. Accessed date: 20 July 2021 (https://doi.org/10.5061/dryad.6q573n5zz).

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
