# Peer review of "Olfactory Learning Supports an Adaptive Sugar-Aversion Gustatory Phenotype in the German Cockroach"

_insects, 2021, doi:10.3390/insects12080724_

Round 1
Reviewer 1 Report
Katsumata and Schal investigated the role of associative learning in the aversive response of GA German cockroaches. The study was conducted using a series of 4 bioassays, in which they demonstrate that the aversive response of GA cockroaches may be paired with discrete olfactory stimuli. This is important because bait formulations for the management of urban cockroach pests, uses olfactory attractants that may lose efficacy quickly if the insect learns to associate the bait with the phagostimulant (glucose). Further, the authors demonstrate that the learned association is maintained for up to 3 days before being extinguished.
Let me just say that I rather enjoyed this manuscript. It was well conceived and very well written, the figures were great, and I thought the discussion was thoughtful. Overall, I think this manuscript is a great contribution and demonstrates learning in GA cockroaches quite well. However, at times the manuscript was verbose and the amount of detail included made it difficult to digest the more salient points. I think it could be improved by a little careful editing to make sure it is focused and direct. This will help the reader fully appreciate the work you did without getting lost in the weeds, so to speak. I have included a few line comments below that address this and a few other minor points.
Figure 1 caption: This caption is quite long. I would remove much of the methods information and instead focus on walking the reader through the diagram.
Figure 1 A: Could you include the measurements of the olfactometer here on the figure?
Figure 1B: Would you consider moving the Conditioning labels portion of the figure to a separate table? Having it off to the side like this, de-emphasizes the importance of this information.
Figure 1C: I think you should reconsider this diagram. I think for most entomologists, this type of schedule diagram is confusing. You would probably be better off making a table. Also, you refer to scotophase repeatedly, but really all this means is that you did your testing during the day. A single mention in the text of the methods would suffice. I would really parse back the language to make it clearer what you did and avoid overcomplicating things.
Ln 55: There are some other really great examples of learning in pest species you could cite and discuss to put your work into the greater context of how insect learning may affect pest management. Here are a few citations to consider, first two are on Rodnius Prolixus, the vector of chagas disease; and second two are on Diaphorina citri an agricultural pest.
Vinauger, C., Buratti, L., & Lazzari, C. R. (2011). Learning the way to blood: first evidence of dual olfactory conditioning in a blood-sucking insect, Rhodnius prolixus. I. Appetitive learning. Journal of Experimental Biology, 214(18), 3032-3038.
Vinauger, C., Lallement, H., & Lazzari, C. R. (2013). Learning and memory in Rhodnius prolixus: habituation and aversive operant conditioning of the proboscis extension response. Journal of Experimental Biology, 216(5), 892-900.
Stockton, D. G., Martini, X., Patt, J. M., & Stelinski, L. L. (2016). The influence of learning on host plant preference in a significant phytopathogen vector, Diaphorina citri. PLoS One, 11(3), e0149815.
Stockton, D. G., Pescitelli, L. E., Ebert, T. A., Martini, X., & Stelinski, L. L. (2017). Induced preference improves offspring fitness in a phytopathogen vector. Environmental entomology, 46(5), 1090-1097.
Ln 179: Could you address why males only were used?
Lns 220-225: Conventionally, the statistical analysis is reported in a separate section. I would consider making this change.
Ln 224: Due to the high number of analyses on single data sets, it would seem reasonable to apply Bonferroni corrections to your alpha level in some cases. Was this considered? If it was and the authors decided against it, perhaps they could include an explanation in the statistics section.
Ln 224: Please include the statistical software used for the analyses.
Figure 2: While I greatly appreciate the amount of work it took to collect these data, I have to admit that I’m not sure how these data move the story forward. I would consider moving this to the supplemental data, especially given how much other data you have. Instead, I would focus on the Figure 3 for bioassay 1.
Ln 364: You say here that you compared the overall responses of WT and GA flies using Chi Sq. I assume a Chi Sq test of independence, like a contingency table? I would elaborate more on the types of statistical comparisons you made. I would also strongly recommend using a more robust approach to your analysis like logistic regression to make comparisons between groups. It would give you more a lot more information and you can include genotype, and odorant concentration in a single model. Then you could use posthoc testing for pairwise comparisons.
Ln 390-393: I would be careful here. Only use the results to report the results. This is discussion. The same on lns 410-412.
Ln 468: I would edit this by removing “in places” from the sentence.
Lns 471-490: I would remove all of this text as it is not a good fit for the discussion. It may also be redundant with the intro. You have so much information to cover, that these additional background points make the discussion much too lengthy. Instead, summarize why you did what you did in 1-2 sentences and then begin interpreting your results.
Ln 510-513: Be careful not to overextend on your interpretation. I’m not sure you can make this recommendation just yet based on your data.
Ln 516-600: I would consider reworking this section to tighten it up and make it more focused. It reads more like a literature review than a discussion and it makes it difficult to discern the important points. Try to get this down to 1-2 paragraphs max. You can summarize the important points as follows:
- “odor preferences increased when associated with rewarding…tastants and decreased when associated with punishing tastants.” Ln 549
- “one hour of operant conditioning using rewarding tastants was sufficient to increase their odor preference 24 hours later.” Ln 552
- “…training using tastants as punishment required 3 1 hr conditioning sessions per day to modify the odor preferences.” Ln 564
- “Different numbers of conditioning sessions were required to impost modifications of odor preferences by rewarding and punishing tastants…” Ln 577 (To be honest, I think you could easily combine points 2-4 here in a single paragraph).
- “preference for an odor was that was associated with a rewarding tastant was enhanced when the cockroaches formed two types of memories by receiving two pairs of odor + tastant combindations.” Ln 592
- “such a memory was retained for at least 3 days.” Ln 600
Ln 603: Is this remarkable?
Ln 612-615: Again, be careful to not make over generous recommendations. I’m not sure you can make this statement just yet. You haven’t tested this hypothesis.
Author Response
Reviewer 1
Katsumata and Schal investigated the role of associative learning in the aversive response of GA German cockroaches. The study was conducted using a series of 4 bioassays, in which they demonstrate that the aversive response of GA cockroaches may be paired with discrete olfactory stimuli. This is important because bait formulations for the management of urban cockroach pests, uses olfactory attractants that may lose efficacy quickly if the insect learns to associate the bait with the phagostimulant (glucose). Further, the authors demonstrate that the learned association is maintained for up to 3 days before being extinguished. Let me just say that I rather enjoyed this manuscript. It was well conceived and very well written, the figures were great, and I thought the discussion was thoughtful. Overall, I think this manuscript is a great contribution and demonstrates learning in GA cockroaches quite well. However, at times the manuscript was verbose and the amount of detail included made it difficult to digest the more salient points. I think it could be improved by a little careful editing to make sure it is focused and direct. This will help the reader fully appreciate the work you did without getting lost in the weeds, so to speak. I have included a few line comments below that address this and a few other minor points.
Let me just say that I rather enjoyed this manuscript. It was well conceived and very well written, the figures were great, and I thought the discussion was thoughtful. Overall, I think this manuscript is a great contribution and demonstrates learning in GA cockroaches quite well. However, at times the manuscript was verbose and the amount of detail included made it difficult to digest the more salient points. I think it could be improved by a little careful editing to make sure it is focused and direct. This will help the reader fully appreciate the work you did without getting lost in the weeds, so to speak. I have included a few line comments below that address this and a few other minor points.
Thank you for the very useful suggestions. We reorganized Fig. 1 and the Discussion to make them simpler for readers to follow. We added two references, as suggested, and made a new Table 1 to support Fig. 1. However, we left Fig. 2 in the main text because we thought it was an important component to show the basic olfactory behavior of WT and GA.
Figure 1 caption: This caption is quite long. I would remove much of the methods information and instead focus on walking the reader through the diagram.
We shortened the caption, as suggested.
Figure 1 A: Could you include the measurements of the olfactometer here on the figure?
We included these details in Fig. 1A.
Figure 1B: Would you consider moving the Conditioning labels portion of the figure to a separate table? Having it off to the side like this, de-emphasizes the importance of this information.
We added Table 1 to explain the symbols of conditioning.
Figure 1C: I think you should reconsider this diagram. I think for most entomologists, this type of schedule diagram is confusing. You would probably be better off making a table. Also, you refer to scotophase repeatedly, but really all this means is that you did your testing during the day. A single mention in the text of the methods would suffice. I would really parse back the language to make it clearer what you did and avoid overcomplicating things.
We made Table 1 to explain the schedule of experiments and removed several “scotophase” from the main text.
Ln 55: There are some other really great examples of learning in pest species you could cite and discuss to put your work into the greater context of how insect learning may affect pest management. Here are a few citations to consider, first two are on Rodnius Prolixus, the vector of chagas disease; and second two are on Diaphorina citri an agricultural pest.
Vinauger, C., Buratti, L., & Lazzari, C. R. (2011). Learning the way to blood: first evidence of dual olfactory conditioning in a blood-sucking insect, Rhodnius prolixus. I. Appetitive learning. Journal of Experimental Biology, 214(18), 3032-3038.
Vinauger, C., Lallement, H., & Lazzari, C. R. (2013). Learning and memory in Rhodnius prolixus: habituation and aversive operant conditioning of the proboscis extension response. Journal of Experimental Biology, 216(5), 892-900.
Stockton, D. G., Martini, X., Patt, J. M., & Stelinski, L. L. (2016). The influence of learning on host plant preference in a significant phytopathogen vector, Diaphorina citri. PLoS One, 11(3), e0149815.
Stockton, D. G., Pescitelli, L. E., Ebert, T. A., Martini, X., & Stelinski, L. L. (2017). Induced preference improves offspring fitness in a phytopathogen vector. Environmental entomology, 46(5), 1090-1097.
We added the suggested papers as Ref #29 (Vinauger et al., 2013) and #30 (Stockton et al., 2016).
Ln 179: Could you address why males only were used?
We addressed this in Materials and Method 2.1.
Lns 220-225: Conventionally, the statistical analysis is reported in a separate section. I would consider making this change.
We made a new section to explain the data analysis as Materials and Methods 2.12
Ln 224: Due to the high number of analyses on single data sets, it would seem reasonable to apply Bonferroni corrections to your alpha level in some cases. Was this considered? If it was and the authors decided against it, perhaps they could include an explanation in the statistics section.
We re-tested all data by Chi-square test with Bonferroni corrections to avoid making multiple statistical tests. Although the results were the same as the previous results by Tukey’s WSD, we now employ the Chi-square test with Bonferroni correction. We describe the method in Materials and Method 2.12.
Ln 224: Please include the statistical software used for the analyses.
We included it in Materials and Method 2.12.
Figure 2: While I greatly appreciate the amount of work it took to collect these data, I have to admit that I’m not sure how these data move the story forward. I would consider moving this to the supplemental data, especially given how much other data you have. Instead, I would focus on the Figure 3 for bioassay
We agree that Fig 2 is not central to our story. However, we decided to keep Figure 2 in the main text, because it is important to show the olfactory response of two strains are almost the same for various odor compounds. Moreover, supplementary figures are often lost in the literature and we think this figure is important for researchers investigating chemical ecology of Blattella in general.
Ln 364: You say here that you compared the overall responses of WT and GA flies using Chi Sq. I assume a Chi Sq test of independence, like a contingency table? I would elaborate more on the types of statistical comparisons you made. I would also strongly recommend using a more robust approach to your analysis like logistic regression to make comparisons between groups. It would give you more a lot more information and you can include genotype, and odorant concentration in a single model. Then you could use posthoc testing for pairwise comparisons.
To compare the % responder between WT and GA in Fig. 2, 4, 5 and 6, we re-tested the data by Chi-square with Bonferroni corrections. We indicated this in Materials and Method 2.12. While we appreciate that a unified model would be statistically appropriate, our design was based on successive hypotheses that were not tested concurrently. Therefore, we opted to retain the Chi-square tests of independence.
Ln 390-393: I would be careful here. Only use the results to report the results. This is discussion. The same on lns 410-412.
We removed these sentences.
Ln 468: I would edit this by removing “in places” from the sentence.
Removed.
Lns 471-490: I would remove all of this text as it is not a good fit for the discussion. It may also be redundant with the intro. You have so much information to cover, that these additional background points make the discussion much too lengthy. Instead, summarize why you did what you did in 1-2 sentences and then begin interpreting your results.
We removed these sentences and rephrased what we did in this study.
Ln 510-513: Be careful not to overextend on your interpretation. I’m not sure you can make this recommendation just yet based on your data.
We removed these sentences.
Ln 516-600: I would consider reworking this section to tighten it up and make it more focused. It reads more like a literature review than a discussion and it makes it difficult to discern the important points. Try to get this down to 1-2 paragraphs max. You can summarize the important points as follows:
- “odor preferences increased when associated with rewarding…tastants and decreased when associated with punishing tastants.” Ln 549
- “one hour of operant conditioning using rewarding tastants was sufficient to increase their odor preference 24 hours later.” Ln 552
- “…training using tastants as punishment required 3 1 hr conditioning sessions per day to modify the odor preferences.” Ln 564
- “Different numbers of conditioning sessions were required to impost modifications of odor preferences by rewarding and punishing tastants…” Ln 577 (To be honest, I think you could easily combine points 2-4 here in a single paragraph).
- “preference for an odor was that was associated with a rewarding tastant was enhanced when the cockroaches formed two types of memories by receiving two pairs of odor + tastant combindations.” Ln 592
- “such a memory was retained for at least 3 days.” Ln 600
We re-organized this whole section based on thes suggestions.
Ln 603: Is this remarkable?
We inserted “important” instead of “remarkable”
Ln 612-615: Again, be careful to not make over generous recommendations. I’m not sure you can make this statement just yet. You haven’t tested this hypothesis.
We removed “and faster evolution of behavioral and insecticide resistance among survivors”.

Reviewer 2 Report
Well-written and designed paper. Very thorough and has excellent potential for applied applications.
Author Response
Reviewer 2
Well-written and designed paper. Very thorough and has excellent potential for applied applications.
Thank you so much for the supportive comment.
Reviewer 3 Report
Comments for authors
Wada-Katsumata and Schal developed the operant olfactory conditioning procedure using the German cockroaches and revealed that the cockroach has an excellent capability of olfactory learning and memory. Especially, wild-type cockroaches receive glucose as a reward in olfactory conditioning, whereas glucose-aversion phenotype cockroaches perceive it as a punishment. The manuscript indicates peripheral simple rejection to tastants affect olfactory memory formation, and resulting dramatical changes of odor-guiding behaviors. In addition, because aversive memory to glucose for GA cockroaches enhance behavioral resistance of sugar-containing toxic bite, the manuscript helps to develop new strategies for controlling the pest insect. Conditioning procedures are well organized and obtained results are clear and very interesting. However, I have some methodological issues. This manuscript must be interesting for general readers of Insects and I would like to see the article in Insect after revisions.
Major comments:
In conditioning, the author used ‘caffeine’ as an aversive unconditioned stimulus. Why did the authors select the caffeine as US? As mentioned in the manuscript, saline solution often used as an aversive US in the olfactory conditioning experiments of cockroaches. In associative learnings using holometabolous and hemimetabolous insects, it has been suggested caffeine directly affects central brain neurons and enhance the olfactory learning (Wright et al., 2013, Science 339, 1202-1204). In hemimetabolous insect crickets, very small amount of caffeine injected into the hemolymph exhibits significant effects on long-term memory formation (Sugimachi et al., 2016, Zoolog Sci 33, 513-519.). If cockroaches did not intake caffeine during the conditioning, the chemical may affect the conditioning. For example, in honeybees, caffeine placed on the thorax also enhance the conditioning effects (Si et al., 2005, Pharmacol Biochem Behav 82, 664-672.). In other word, glucose for GA cockroaches may not affects central brain neurons during conditioning. Are there any differences of conditioning effects between two aversive USs? It seems that caffeine did not affect olfactory conditioning in this study. Nevertheless, the authors have to describe the effects of caffeine on olfactory conditioning in Introduction or Discussion.
In memory retention experiments, the author described that olfactory memory was tested at 2, 3 and 5 days after conditioning throughout the manuscript. However, scheme of Fig. 1C looks that olfactory memory tested 2, 3 or 5 days later. Please clarify the experimental protocol of the retention experiment. In this study, long-term memory formed by operant conditioning did not retain at 5 days after conditioning. It looks very interesting because long-term memory is generally retained during life period unless receiving the reversal conditioning or memory extinction (Bitterman et al., 1983, J Comp Psychol 97, 107-119; Matsumoto and Mizunami, 2002, J Comp Physiol A 188, 295-299; Sakura and Mizunami, 2003). If the conditioned cockroach receives repeated test sessions in which conditioned odors presented alone, the long-term olfactory memory will be extinct as shown in Figure 6. In addition, if individual cockroach received preference tested at 2, 3 and 5 days after conditioning, odor preferences at 1 day after conditioning should be omitted from Figure 6.
Minor comments:
L69
underlying (GA) is > underlying GA is
L72
Glucose-aversion > GA
L120
The tubes were exhausted > the air was exhausted
L124 and L139
The authors described that “quiescent insects” were used in bioassay. Is it true?
L171-175
In 2.5. chemicals, there are no information of chemicals used as tastants and solvents. In addition, it is better to describe the purities of chemicals.
L 264
US (either vanilla or chocolate) > US (either fructose, glucose or caffeine)?
I cannot follow the sentence.
Page 7, last paragraph
The authors described that 2,4,5-trimethylthiazole were not discriminated from the solvent control at any of concentrations (Lines 304-306). But, in a following sentence, they also described that 2,4,5-trimethylthiazole acted as attractant at a modulate concentration. The two sentences seem to be contradicting each other.
Line 328-329
The authors sometime described that the commercial extracts containing multiple odor compounds were more attractive than single compounds. However, the value of % responding of >0.1 µg of vanillin is higher than that of the vanilla extract.
Line 383
‘Fructose + Vanilla’ > ‘Vanilla + Fructose’
Page 14, third paragraph
The authors suggested that blended odors enable greater resolution and discrimination of odor sources. I guess vanillin is a major component of vanilla extract. To compare the odor preference between single and blended odors, it needs to denote compositions of vanilla and chocolate extracts used in this study.
Line 520
(mushroom body, and lateral horn) > (mushroom body and lateral horn)
Line 611
suggests that that the GA > suggests that the GA
Figure 1C
Change “20 pm” to “8 pm”.
Table S1
Did the author test 330 cockroaches in the preference test for 0.01 μg of beta-caryophyllene?
Author Response
Reviewer 3
Wada-Katsumata and Schal developed the operant olfactory conditioning procedure using the German cockroaches and revealed that the cockroach has an excellent capability of olfactory learning and memory. Especially, wild-type cockroaches receive glucose as a reward in olfactory conditioning, whereas glucose-aversion phenotype cockroaches perceive it as a punishment. The manuscript indicates peripheral simple rejection to tastants affect olfactory memory formation, and resulting dramatical changes of odor-guiding behaviors. In addition, because aversive memory to glucose for GA cockroaches enhance behavioral resistance of sugar-containing toxic bite, the manuscript helps to develop new strategies for controlling the pest insect. Conditioning procedures are well organized and obtained results are clear and very interesting. However, I have some methodological issues. This manuscript must be interesting for general readers of Insects and I would like to see the article in Insect after revisions.
Major comments:
In conditioning, the author used ‘caffeine’ as an aversive unconditioned stimulus. Why did the authors select the caffeine as US? As mentioned in the manuscript, saline solution often used as an aversive US in the olfactory conditioning experiments of cockroaches. In associative learnings using holometabolous and hemimetabolous insects, it has been suggested caffeine directly affects central brain neurons and enhance the olfactory learning (Wright et al., 2013, Science 339, 1202-1204). In hemimetabolous insect crickets, very small amount of caffeine injected into the hemolymph exhibits significant effects on long-term memory formation (Sugimachi et al., 2016, Zoolog Sci 33, 513-519.). If cockroaches did not intake caffeine during the conditioning, the chemical may affect the conditioning. For example, in honeybees, caffeine placed on the thorax also enhance the conditioning effects (Si et al., 2005, Pharmacol Biochem Behav 82, 664-672.). In other word, glucose for GA cockroaches may not affects central brain neurons during conditioning. Are there any differences of conditioning effects between two aversive USs? It seems that caffeine did not affect olfactory conditioning in this study. Nevertheless, the authors have to describe the effects of caffeine on olfactory conditioning in Introduction or Discussion.
We elaborate on caffein’s sensory properties in Materials and Methods 2.4 and Discussion 4.2. Additionally, we refer to the two papers (Wright et al., 2013 and Si et al., 2005) that deal with caffeine’s role in learning.
In memory retention experiments, the author described that olfactory memory was tested at 2, 3 and 5 days after conditioning throughout the manuscript. However, scheme of Fig. 1C looks that olfactory memory tested 2, 3 or 5 days later. Please clarify the experimental protocol of the retention experiment. In this study, long-term memory formed by operant conditioning did not retain at 5 days after conditioning. It looks very interesting because long-term memory is generally retained during life period unless receiving the reversal conditioning or memory extinction (Bitterman et al., 1983, J Comp Psychol 97, 107-119; Matsumoto and Mizunami, 2002, J Comp Physiol A 188, 295-299; Sakura and Mizunami, 2003). If the conditioned cockroach receives repeated test sessions in which conditioned odors presented alone, the long-term olfactory memory will be extinct as shown in Figure 6. In addition, if individual cockroach received preference tested at 2, 3 and 5 days after conditioning, odor preferences at 1 day after conditioning should be omitted from Figure 6.
Thank you for this important suggestion. We corrected this to “Conditioned preference bioassays with vanilla and chocolate were conducted 2, 3 or 5 days later”, and mentioned it again in Table 1 which we added to the main text.
Minor comments:
L69
underlying (GA) is > underlying GA is
Corrected.
L72
Glucose-aversion > GA
Corrected.
L120
The tubes were exhausted > the air was exhausted
Corrected.
L124 and L139
The authors described that “quiescent insects” were used in bioassay. Is it true?
It is true. When the insects calmed down and stopped waking in the chamber, we started the bioassay.
L171-175
In 2.5. chemicals, there are no information of chemicals used as tastants and solvents. In addition, it is better to describe the purities of chemicals.
We added the information about solvents and tastants in Materials and Method 2. 5.
L 264
US (either vanilla or chocolate) > US (either fructose, glucose or caffeine)?
I cannot follow the sentence.
We removed the sentence “namely, the CS (either vanilla or chocolate) and US (either vanilla or chocolate)” to avoid confusion.
Page 7, last paragraph
The authors described that 2,4,5-trimethylthiazole were not discriminated from the solvent control at any of concentrations (Lines 304-306). But, in a following sentence, they also described that 2,4,5-trimethylthiazole acted as attractant at a modulate concentration. The two sentences seem to be contradicting each other.
We rephrased these sentences.
Line 328-329
The authors sometime described that the commercial extracts containing multiple odor compounds were more attractive than single compounds. However, the value of % responding of >0.1 µg of vanillin is higher than that of the vanilla extract.
We removed the sentence “These results indicate that in our olfactometers, the commercial extracts containing multiple odor compounds were more attractive than single compounds.”
Line 383
‘Fructose + Vanilla’ > ‘Vanilla + Fructose’
Corrected.
Page 14, third paragraph
The authors suggested that blended odors enable greater resolution and discrimination of odor sources. I guess vanillin is a major component of vanilla extract. To compare the odor preference between single and blended odors, it needs to denote compositions of vanilla and chocolate extracts used in this study.
We included the information on the ingredients of commercial products in Materials and Methods 2. 5. However, we could not obtain details of components from the data sheets of commercial products. Thus, we rephrased the sentence in Results 4.1. to “Considering that the commercial extracts contain multiple compounds from natural materials, these findings suggest that blended odors may enable greater resolution and discrimination of odor sources.“.
Line 520
(mushroom body, and lateral horn) > (mushroom body and lateral horn)
Corrected.
Line 611
suggests that that the GA > suggests that the GA
Corrected.
Figure 1C
Change “20 pm” to “8 pm”.
We removed Fig. 1C and added a new Table 1 to explain the experimental design.
Table S1
Did the author test 330 cockroaches in the preference test for 0.01 μg of beta-caryophyllene?
Thank you so much for the point. It was a mistype. We tested 30 males.
